# Uncovering Psychedelics: From Neural Circuits to Therapeutic Applications

**DOI:** 10.3390/ph18010130

**Published:** 2025-01-19

**Authors:** Alice Melani, Marco Bonaso, Letizia Biso, Benedetta Zucchini, Ciro Conversano, Marco Scarselli

**Affiliations:** 1Department of Biology, University of Pisa, 56126 Pisa, Italy; a.melani7@studenti.unipi.it; 2BIO@SNS Lab, Scuola Normale Superiore, 56126 Pisa, Italy; 3Department of Translational Research and New Surgical and Medical Technologies in Medicine and Surgery, University of Pisa, 56126 Pisa, Italy; m.bonaso@studenti.unipi.it (M.B.); l.biso@studenti.unipi.it (L.B.); b.zucchini@studenti.unipi.it (B.Z.); 4Department of Surgical, Medical and Molecular Pathology, and Critical Care Medicine, University of Pisa, 56126 Pisa, Italy; ciro.conversano@unipi.it

**Keywords:** psychedelics, DMN, CSTC, REBUS, psilocybin, MDMA, LSD, TRD, GAD, PTSD

## Abstract

Psychedelics, historically celebrated for their cultural and spiritual significance, have emerged as potential breakthrough therapeutic agents due to their profound effects on consciousness, emotional processing, mood, and neural plasticity. This review explores the mechanisms underlying psychedelics’ effects, focusing on their ability to modulate brain connectivity and neural circuit activity, including the default mode network (DMN), cortico-striatal thalamo-cortical (CSTC) loops, and the relaxed beliefs under psychedelics (REBUS) model. Advanced neuroimaging techniques reveal psychedelics’ capacity to enhance functional connectivity between sensory cerebral areas while reducing the connections between associative brain areas, decreasing the rigidity and rendering the brain more plastic and susceptible to external changings, offering insights into their therapeutic outcome. The most relevant clinical trials of 3,4-methylenedioxymethamphetamine (MDMA), psilocybin, and lysergic acid diethylamide (LSD) demonstrate significant efficacy in treating treatment-resistant psychiatric conditions such as post-traumatic stress disorder (PTSD), depression, and anxiety, with favorable safety profiles. Despite these advancements, critical gaps remain in linking psychedelics’ molecular actions to their clinical efficacy. This review highlights the need for further research to integrate mechanistic insights and optimize psychedelics as tools for both therapy and understanding human cognition.

## 1. Introduction

Coined in the 1950s, the term “psychedelic” comes from the Greek words ψυχή (psyche, ’soul, mind’) and δηλοῦν (deloun, ‘to manifest’) and is used to describe the subjective effects of these substances, highlighting their ability to alter perception, state of consciousness, introspection, empathy, cognitive processes, and mood [1]. They are generally considered safe and they should not lead to dependence or addiction. The several ongoing clinical trials on psychedelic-based drugs have certainly helped to bring psychedelics back to the center of scientific debate, stimulating the clarification of their mechanisms of action and enlightening their potential therapeutic use [1]. Psychedelics have long captivated human curiosity, not only for their cultural and spiritual significance but also for their profound effects on consciousness, emotions, and cognitive functions, together with an increase in emotional processing, introspection, and a sense of interconnectedness. Their intake can lead to unique transformative experiences marked by a peculiar spectrum of effects, including perceptive, emotional, cognitive, behavioral, and psychophysiological alterations [2]. Historically, with the rise in recreational use in the counterculture movements of the 1960s, psychedelics came under increasing scrutiny and were classified as illegal substances. This ban stifled research and relegated psychedelics to the margins of society and science. Only in recent years has there been a resurgence of interest in psychedelics, driven by a growing body of research highlighting their therapeutic potential, named the psychedelic renaissance [3].

As we could appreciate from the evidence presented in this review, many clinical studies have begun to reveal that, when used in controlled settings, psychedelics can have significant positive effects on various mental health conditions [4]. Among all, research has shown that psychedelics can induce altered states of consciousness, leading to enhanced emotional processing, increased introspection and empathy, a sense of interconnectedness, and reduction in negative bias and social withdrawal [5]. These experiences may provide new avenues for treating a range of psychopathological conditions, including depression, anxiety, post-traumatic stress disorder (PTSD), and substance use disorders (SUDs) [6]. Clinical studies have reported promising results, demonstrating significant improvements in symptoms and quality of life for patients undergoing psychedelic-assisted therapy [7]. Notably, recent findings expand this discourse by exploring psychedelics’ impact on under-researched domains such as sexual function, states of consciousness, and the neurobiology underlying psychiatric disorders [8,9,10].

While significant progress has been made, critical gaps remain for linking psychedelics’ molecular mechanisms to their clinical characteristics. In particular, it is essential to understand the contribution of the psychedelic experience and the increased neuroplasticity for their overall clinical efficacy, and how these two elements are connected to each other. Furthermore, understanding the changes in neural circuits underlying the neuroplastic effects of psychedelics is another critical aspect to elucidate their properties concerning cognitive functions, states of consciousness, and changes in mood. On this topic, the default mode network (DMN), the executive control network (ECN), the salience network (SN), and two models, i.e., cortico-striatal thalamo-cortical (CSTC) and the relaxed beliefs under psychedelics (REBUS), have received a lot of attention [11,12,13]. In addition, future studies should integrate such emerging knowledge critically, advancing psychedelics’ research not only as therapeutic agents but also as tools to unravel the brain’s deepest functions.

Based on this premise, the purpose of this review is to provide a state of the art of psychedelics’ research, not only elucidating the mechanisms and the neural circuits underlying their effects, but also updating their clinical use for different psychiatric disorders. Specifically, the most advanced clinical trials of psychedelics such as 3,4-methylenedioxymethamphetamine (MDMA), psilocybin, and lysergic acid diethylamide (LSD) for treating PTSD, treatment-resistant depression (TRD), and generalized anxiety disorder (GAD) will be overviewed.

## 2. Molecular and Cellular Targets of Psychedelics

Various pieces of evidence, based on ex vivo/in vitro binding assays, have demonstrated the receptor specificities of psychedelic substances. According to this principle, psychedelics can be divided into two different classes: serotonergic psychedelics and monoaminergic psychedelics. Serotonergic psychedelics, such as psilocybin, are mainly partial agonists at 5HT2AR; thus, other 5HTR such as 5HT1AR can be involved in their mechanism. In addition, compounds such as LSD can also be partial agonists at dopaminergic D1/D2 receptors [1]. The monoaminergic psychedelics, to which belongs MDMA, consist of inhibitors of monoamine reuptake and inhibitors of VMAT2. Furthermore, another substance that is used to perform experiments on psychedelics’ effects is 2,5-dimethoxy-4-iodoamphetamine (DOI), which is a synthetic amphetamine whose effects are compared to the ones of LSD even though they have notable differences [14,15]. Besides these two main classes, we should also mention the glutamatergic NMDA receptor antagonists ketamine and Esketamine, where the last one has been recently approved for treatment-resistant depression. However, ketamine and Esketamine at the low doses used to treat depression mainly induce a dissociative effect rather than a psychedelic one [16]. Ongoing studies are then arguing whether this dissociative role of ketamine is actually necessary for its antidepressant effects. According to this, a recent review shows that a poor correlation exists between dissociative symptoms and therapeutic outcomes. Instead, it is proposed that the psychedelic effects induced by slightly higher doses of ketamine are the ones able to promote changes within the highest levels of the cerebral hierarchy, leading to alterations in self-representation [17].

5-HT2ARs are coupled to Gq protein, but they also have biased activity towards the β-arrestin-dependent transduction pathway [18,19,20]. The canonical process through the activation of PLCγ initiates the release of intracellular calcium through inositol triphosphate (IP3) and causes the activation of protein kinase C (PKC). Furthermore, the activation of 5-HT2AR has been recently demonstrated to activate β-arrestin via PI3K and AKT, with a different degree of recruitment. In a recent study published in *Nature*, Wallach et al. showed that the effectiveness of recruitment of the 5-HT2A-Gq pathway—and not 5-HT2A-β-arrestin2—predicts psychedelic potential, assessed using the magnitude of the mouse head-twitch response (HTR). Furthermore, they even elucidated that the disruption of Gq-PLC signaling attenuates HTR and that, in turn, a threshold level of Gq activation is required to induce psychedelic-like effects, consistent with the fact that some partial agonists of 5-HT2A (e.g., lysuride) are non-psychedelic. However, the biased agonism of psychedelics at 5-HT2AR requires further examination [19].

For what concerns serotonergic psychedelics, which are the main subject of the present review, the activation of 5-HT2A receptors located in cortical and subcortical structures is a unifying mechanism through which psychedelics mediate their behavioral and psychological effects in animals, including humans. Pharmacological studies have shown that the mouse head-twitch response (HTR) paradigm in rodents, a behavioral proxy for the effects of hallucinogens on humans, depends on the stimulation of the 5-HT2A receptor but not on the stimulation of the 5-HT2C or 5-HT2B receptor [21]. The relevance of 5-HT2A has also been confirmed in humans, where the 5-HT2A receptor antagonist ketanserin abolishes practically all the subjective effects of psilocybin, LSD, and DMT [22]. Through immunohistochemistry or autoradiography, we can observe that 5-HT2ARs are mainly localized in the apical dendrites of layer 5 pyramidal neurons (L5p) of the neocortex, with a significant concentration at the level of the prefrontal cortex (PFC); then, they are localized in presynaptic afferences of thalamo-cortical projections and on GABAergic neurons of cortical and subcortical structures [23]. These areas are associated with functions such as visual perceptions and attention, which contribute to the wide and diverse effects of hallucinogens on emotional, perceptual, and cognitive processes.

In addition, given the antidepressant effect induced by psychedelics in mouse models, assessed through a forced swimming test or fear extinction learning, it has been demonstrated that serotonergic psychedelics are able to induce neuroplasticity, similarly to the well-known antidepressant ketamine. The demonstration of the neuroplastic effects of serotonergic psychedelics has allowed the definition of psychoplastogens, from the Greek “psych” (mind), “plast” (modeled), and “gen” (produce) [24].

Recent research conducted by Moliner et al. in mice also suggests that the therapeutic effects of psychedelics may be more closely linked to the activation of the BDNF-TrkB signaling pathway. In particular, some experiments showed that psychedelics, for example, LSD, are actually positive allosteric ligands of TrkB and are able to bind to a site located at the intersection of the transmembrane domains of TrkB, facilitating its dimerization and making the receptor more prone to BDNF binding. At the macroscopic level, this causes the promotion of synaptic plasticity in an activity-dependent manner, the increase in spinogenesis and dendritogenesis, and the facilitation of the survival of newborn neurons in the hippocampus and their incorporation in the dentate gyrus (DG) and promotes the rewiring of neuronal networks, resulting in important behavioral effects that can be exploited therapeutically [25]. Lastly, this study showed that the psychedelic effect depends on the binding to 5HT2A-R, while the induction of neuroplasticity, at the basis of the therapeutic effects of these substances, can be induced by the activation of both 5HT2A-R and the TrkB pathway. The development of new psychedelic-based drugs more selective for TrkB would therefore allow us to obtain neuroplastic effects by preventing the induction of the psychedelic effect. The relevance of the neurotrophic effect versus the psychedelic effect for the therapeutic success of the serotonergic psychedelics is still under intense scientific scrutiny, which probably will be solved by the discovery of more selective drugs [25].

Regarding monoaminergic psychedelics, MDMA is a psychoactive drug with powerful prosocial effects. While MDMA is sometimes termed an “empathogen,” empirical studies have struggled to clearly demonstrate these effects. MDMA’s mechanism of action consists in reverting the reuptake of monoamines (5HT, DA, NA), thus increasing their release at a synaptic level. This macroscopic molecular effect is guaranteed by the interaction of MDMA with various targets. Firstly, MDMA is a substrate of monoamine transporters (SERT, DAT, and NAT), which provide its transport inside the cell, where it can promote the inhibition of vesicular monoamine transporter 2 (VMAT2), together with the agonistic activation of TAAR1 [15,26,27]. In a pioneering article of 1992, Heifets et al. demonstrated that MDMA exerts actions at serotonin transporters present in the plasma membrane and secretory vesicles, from which MDMA stimulates serotonin efflux. At the level of the secretory vesicles, which contain the vesicular biogenic amine transporter (VMAT2), MDMA inhibits ATP-dependent serotonin accumulation and stimulates efflux, thus increasing serotonin release at a synaptic level [2]. The activation of the trace amine-associated receptor (TAAR1) stimulates PKA and PKC pathways, leading the downstream phosphorylation of the monoamine transporters, causing the ultimate inversion of the monoamine transport that, once accumulated in the cytosol, can be released at the level of the synaptic cleft, increasing monoaminergic tone [28]. Then, since MDMA’s affinity is 10 times higher for SERT with respect to the other transporters, its main effect is to enhance serotoninergic tone. Regarding MDMA’s behavioral effects, many previous studies have demonstrated MDMA’s ability to increase social preference and interest in rodents, but the paper of Rein et al. is the first one to explore MDMA’s role in promoting empathy-like behaviors, modeled by the development of two specific “social transfer” paradigms in mice: the social transfer of pain and analgesia [29]. For what concerns the social transfer of pain, bystander (BY) mice acquire pain following a brief social interaction with a demonstrator (DEM) mouse experiencing inflammatory pain. On the contrary, in the social transfer of analgesia, BY mice acquire pain relief following a social interaction with a DEM experiencing inflammatory pain and concurrent morphine (MOR) analgesia. The results demonstrated that MDMA enhances both the social transfer of pain and analgesia, consistent with MDMA exhibiting empathogenic properties. Then, by the means of the TRAP (targeted recombination in active populations) technique and optogenetics, they were able to demonstrate that the stimulation of 5-HT release in the Nucleus Accumbens (NAc) alone is sufficient to reproduce the behavioral effects of MDMA; in particular, in a rodent model, researchers observed empathy-like behaviors [29].

These pieces of evidence are consistent with results obtained by Heifets et al.: using complementary genetic and brain region-specific pharmacological manipulations and in vivo calcium imaging, they found that MDMA acts at SERT-containing 5-HT terminals in the NAc. Then, convergent evidence from behavioral pharmacology and electrophysiology experiments further demonstrates that MDMA’s prosocial effect requires the activation of the 5-HT1b receptor in the NAc [2].

On the contrary, MDMA’s nonsocial rewarding effect does not require SERT but rather involves dopaminergic signaling in the NA. Thus, they demonstrated not only that the main region responsible for MDMA’s effects is the NAc, a conserved brain region that regulates appetitive behavior, but even that its prosocial and the rewarding effects are mediated by independent mechanisms. In particular, prosocial effects are mediated by the activation of the serotoninergic system, whereas the rewarding effect requires the activation of the dopaminergic signaling [2].

For what concerns the MDMA therapeutic effect against PTSD, one possible mechanism of action involves its capability to increase brain-derived neurotrophic factor (BDNF) availability in the fear memory learning pathways. Supporting this hypothesis, MDMA has been demonstrated to induce increases in neural plasticity [30,31]. Further studies found that, after MDMA treatment, BDNF protein levels and BDNF mRNA transcripts increase in some areas involved in fear learning, such as the hippocampus, the amygdala, and the vmPFC. In this view, MDMA could reopen a plastic window for the trauma-induced fear memory association, allowing learning extinction and, subsequently, the reconsolidation of the memory to occur [32]. This effect, coupled with MDMA’s prosocial effects, may provide an integrated explanation for its therapeutic action concerning PTSD.

## 3. Circuitries, Connectivity, and Brain Networks Under Psychedelics’ Effect

Until now, there has not been a unified view of psychedelics’ effect on connectivity and brain circuits. Understanding the neural circuit changes under the effect of psychedelics is essential to elucidate their properties concerning cognitive functions, states of consciousness, and changes in mood. Many neuroimaging techniques have been a valuable tool to investigate the impact of psychedelics, and most of them use functional connectivity approaches to investigate changes in between-network and within-network connectivity. Among the various techniques, neuroimaging techniques such as arterial spin labeling (ASL) and blood oxygen level-dependent (BOLD) measures are frequently used [33,34]. In addition, other imaging methods such as emission positron tomography (PET) and the Single-Photon Emission Computed Tomography (SPECT) allow us to evaluate metabolic activity and cerebral blood flow, while functional magnetic resonance (fMRI) provides information about neuronal activity on the basis of the hemodynamic response [35,36,37]. Lastly, electroencephalography (EEG) and magnetoencephalography (MEG) detect, respectively, electric activity and magnetic fields generated by neuronal activity [34,37,38].

By using neuroimaging, one of the pioneering works of Carhart-Harris on psilocybin and LSD revealed that marked changes in brain connectivity strongly correlate with their therapeutic effects. On this subject, several studies have demonstrated that during psychedelic experiences, unique changes in brain connectivity occur depending on the area investigated [33,39]. According to Daws et al., psychedelics enhance the connectivity between cerebral regions that were previously distinct and separated, reflecting the transition from a more segregated brain towards a more interconnected network [40]. For instance, the strongest connections have been found between prefrontal areas, involved in decision and self-control, and sensory areas, such as the primary visual cortex, suggesting a major functional integration and communication between the thalamus and sensory cortical regions during the psychedelic state [41,42]. On the other hand, further studies have demonstrated that psychedelics could significantly reduce activity and connectivity within associative networks [33,41].

Overall, psychedelics increase connections between sensory cerebral areas, while reducing the connections between associative brain areas, decreasing the rigidity of brain networks, and rendering the brain more plastic and susceptible to external changes. Thus, it is conceivable that increased processing of sensory information that is not counterbalanced by associative network integrity may underlie the phenomenology of psychedelics, in particular for what concerns LSD and psilocybin [41,43]. Furthermore, the areas affected by these functional connectivity changes correlate with the pattern of expression of the HTR2A gene, further corroborating the causative relationship between psychedelic cellular targets and their effect on brain connectivity [41].

### 3.1. DMN Model

More specifically, many changes on connectivity at a circuital level concern the default mode network (DMN), an intrinsic brain network composed by four functional hubs: the medial prefrontal cortex (mPFC), the posterior cingulate cortex (PCC), the precuneus, and the angular gyrus. This is consistent with the fact that 5-HT2ARs are expressed most densely in the cortex and especially in high-level association regions, including those belonging to the so-called default mode network [44]. The DMN is one of many resting-state networks (RSNs) and it is active during the resting state or internal mental processing. It is anticorrelated with task-based brain networks, such as the salience network (SN). Gattuso et al. observed that, under psychedelics’ effect, the DMN shows a reduced activity and communication within its components, while increasing the ones with other circuitries [12]. For what concerns the relationship between DMN and other circuitries, it has been observed that there is an increased integration between DMN and the executive control network (ECN), together with a significant reduction in the connectivity within the ECN [45] (Figure 1). The ECN includes the lateral prefrontal cortex, the posterior parietal cortex (PPC), the frontal eye fields (FEFs), and part of the dorsomedial prefrontal cortex (dmPFC) [46]. Contrary to the DMN, this network is most active during cognitive tasks and is implicated in cognitive functioning including attention and working memory. Furthermore, it has even been reported that there is an increased connectivity between DMN and the salience network (SN), which plays an important role in identifying and conferring relevance to significant stimuli coming from the external and the internal world, thus contributing to a more fluid integration between external and internal experiences, balancing them and potentiating cognitive flexibility [47]. Overall, alterations in DMN functional connectivity, consisting of both decreased intraconnectivity (within the DMN brain regions) and increased interconnectivity (between DMN and other cognitive networks), have been implicated in a variety of complex cognitive functions such as self-referencing, memory, and rumination as well as mood conditions like depression. However, the underlying mechanisms are not fully understood, nor revealing a congruent pattern yet. According to Vollenweider et al., the lack of consistency may be because most of the results are based on relatively small sample sizes (usually <20 participants), while the most consistent finding in all of the various studies concerning psychedelic-induced alterations of brain connectivity is the reduced functional connectivity in or between structures of the DMN [1].

### 3.2. CSTC Model

Besides the DMN, there are two other models that have been considered to explain the impact of psychedelics on brain connectivity and cerebral activity: the cortico-striatal thalamo-cortical (CSCT) model and the relaxed beliefs under psychedelics (REBUS) model.

In the CSTC model, the main player is the thalamus, a relay structure consisting of a set of subcortical nuclei that retransmit sensory information from the sensory organs to the cortex, playing an important relay role that allows it to act as a “gate”, limiting the flow of information to upper brain regions [13] (Figure 2). In particular, apart from the reticular nucleus, all thalamic nuclei provide glutamatergic inputs directed to the cortex. The CSTC circuit consists of the pyramidal neurons of the medial prefrontal layer V that project to the GABAergic neurons of the ventral striatum, which in turn inhibit specific GABAergic neurons of the pallidum, which subsequently inhibit some thalamic nuclei that project back to the cortex. Each of these stations expresses 5-HT receptors, especially 5-HT2AR [48]. According to this model, we can understand that the interruption of the activity of these pallido-thalamic GABAergic neurons causes a general disinhibition of the thalamus, in particular of the mediodorsal nucleus, with consequent flooding of the cortex with sensory information. According to this scheme, it has been hypothesized that serotonergic psychedelics are able to reduce the effectiveness of thalamic gating by stimulating 5-HT2A receptors present at various levels of the circuit, increasing sensory flow to the cortex, thus making the brain more responsive to stimuli and inducing the perceptual and cognitive alterations that accompany the psychedelic experience. According to this hypothesis, the disruption of the correct thalamic gating would result in the increase in the sensory perception and dissolution of the ego that occur in psychedelic states. Such a model is supported by behavioral measures of altered sensorimotor gating in humans after the administration of psilocybin, LSD, and ayahuasca [49,50,51]. The model finds support in the results of neuroimaging studies indicating an increase in functional connectivity between the thalamus and sensory cortex in response to LSD, as well as an enhancement of excitatory connectivity from the thalamus to the posterior cingulate cortex (PCC) [41,50]. Together with these pieces of evidence, a peculiar hyperactivity of the prefrontal brain regions was previously observed, marked by the increase in glucose consumption following psilocybin administration [52]. This phenomenon is known as ’hyperfrontality’ and consists of a hypermetabolism of the prefrontal and temporo-medial areas, accompanied by a hypometabolism of the subcortical and occipital brain regions [1,52,53]. According to this model, supported by various pieces of evidence, the reduction in thalamic filtering caused by psychedelics can therefore lead to an increase in sensory processing not counterbalanced by integrative processing in the associative cortices, resulting in amplified or distorted sensory and perceptual experiences. Although it is not known which specific thalamo-cortical projection is linked to each particular feature of the psychedelic state, those who are under the influence of psychedelics report to perceive visual, auditory, and tactile stimuli in an intense and vivid way, combined with alterations in cognitive, emotional, and perceptive functions, contributing to the overall effect of psychedelics on the human experience.

### 3.3. Relaxed Beliefs Under Psychedelics (REBUS) and the Anarchic Brain Model

In the REBUS model, also called the “anarchic brain”, the imbalance between interoceptive and exteroceptive information caused by the reduction in thalamic filtering is compatible with the hypothesis of sensory overflow (bottom-up) and the entropic brain. In practice, it proposes that psychedelics reduce the stability and rigidity of our beliefs and assumptions about the surrounding world and ourselves, and increase our relaxation over prediction. From a psychological point of view, the core idea underpinning this model is that we experience and navigate the world through prior beliefs that we have about it [11].

According to Doss et al., the REBUS model proposes that psychedelics disrupt the hierarchical organization of the brain by reducing the constraints that higher levels place on lower levels. This reduction in top-down control results in the increased influence of low-level prediction errors that, in turn, influence higher levels of processing, which encode or maintain prior beliefs. In this way, a greater influence of incoming stimuli in prediction circuits may allow the expression of more cortical brain states, thus increasing cortical ‘entropy’ and putatively allowing updates or changes to prior beliefs. Preliminary empirical evidence supporting this model, mainly obtained by fMRI, shows that psychedelics such as LSD, psilocybin, DMT, ketamine, and ayahuasca increase the repertoire of different brain states and entropy in brain activity, as a measure of signal diversity detected with MEG [13]. This model is corroborated by much evidence, including reduced electrophysiological responses to the presentation of surprising stimuli under the influence of LSD, as well as reduced top-down and increased bottom-up connectivity between different brain areas such as the thalamus and the PCC. However, this is still controversial since other studies apparently did not confirm these results [54].

Given the DMN’s role in self-referential processing, it has been located on the top of the hierarchical organization of the brain by the REBUS model, considering it as the neurobiological substrate of the ‘ego’ (in the Freudian sense). In this view, the attenuation of DMN function can be interpreted as the result of ‘ego dissolution’ reported after psychedelic administration. In particular, by increasing entropy and decreasing the activity within the components of the DMN, psychedelics are able to increase bottom-up information from the hippocampus and the parahippocampal gyrus, both expressing 5-HT2A receptors together with the entorhinal cortex, attenuating the entorhinal gating of information to the hippocampus [55]. Thus, psychedelics seem to be able to increase bottom-up, over reciprocal top-down, information flow from the hippocampus and parahippocampal gyrus to higher-level cortical areas, especially the DMN, resembling the effects of the thalamic gate disruption. This alteration manifests as an increase in cortical entropy, consistently with many studies that show that psychedelics desynchronize scalp electrophysiological oscillations, especially in lower frequencies, leading to an expansion of possible brain states, as hypothesized by the model [56,57].

Given the centrality of the DMN, controlled by the hippocampus and parahippocampal gyrus, the alteration of these connections, proposed by the REBUS model, could cause and explain a reduction in the self-centered thinking and affect the perception and evaluation of experience, as reported under psychedelics, inducing the generation of states of consciousness in which mental schemes, priors, and beliefs can be temporally relaxed or suspended. Such changes could contribute to psychedelic experiences of connection, introspection, and creativity, although behavioral evidence for this assertion is lacking.

The progressive elucidation of the neural circuits underlying psychedelics’ action is pivotal for advancing more effective therapeutic applications. Many psychiatric conditions, such as depression, anxiety, PTSD, and SUDs, are often rooted in altered brain connectivity, and could be better understood and treated by investigating how psychedelics modulate these circuits [58,59]. A deeper understanding of these mechanisms may pave the way for targeted interventions that leverage psychedelics to restore or enhance functional connectivity in affected networks. For instance, research demonstrates that psychedelics promote neural plasticity through the activation of 5-HT2A receptors, facilitating dendritic growth and synaptic remodeling. These changes may help to restore disrupted neural networks in treatment-resistant depression [60]. Additionally, psilocybin and other psychedelics increase brain entropy and desynchronize neural activity, altering rigid patterns of connectivity often associated with disorders like obsessive–compulsive disorder (OCD) or SUDs [61,62]. The ability of psychedelics to enhance functional connectivity and break maladaptive patterns of brain activity may also explain their efficacy in alleviating PTSD by allowing patients to process traumatic memories in a more adaptive way [63].

## 4. Overview of Major Completed Clinical Trials of MDMA, Psilocybin, and LSD in PTSD, TRD, and GAD

Psychedelics such as MDMA, psilocybin, and LSD are the subject of numerous clinical trials to treat complex and often treatment-resistant psychiatric conditions, such as PTSD, TRD, and GAD. As of October 2024, there were 34 studies registered in the Clinicaltrial.gov database of MDMA for the treatment of PTSD, 21 studies of psilocybin for TRD, 10 studies of psilocybin for PTSD, and a smaller selection of studies of LSD, mostly for GAD [63]. Although only a few of these studies have reached advanced stages, preliminary results show significant symptom reductions and a favorable tolerability profile.

### 4.1. Completed Clinical Trials of MDMA in PTSD

The most advanced studies of MDMA for PTSD are the MAPP1 (ClinicalTrials.gov identifier: NCT03537014) and MAPP2 (ClinicalTrials.gov identifier: NCT04077437) trials, conducted by the Multidisciplinary Association for Psychedelic Studies (MAPS) (Table 1). In these two randomized, double-blind, placebo-controlled, phase 3 studies, MDMA is administered in specific doses (80–120 mg, with a supplemental dose of 40 or 60 mg), during guided psychotherapy sessions. The primary objective was to evaluate the efficacy and safety of this combination therapy for patients with moderate to severe PTSD, many of whom have complicated comorbidities, such as substance abuse, depression, or dissociation [6,64]. The MAPP1 trial included 90 participants, randomized to receive MDMA (*n* = 46) or a placebo (*n* = 44) in combination with three sessions of psychotherapy [65]. The results indicated that patients in the MDMA group experienced a substantial improvement in PTSD symptoms after 18 weeks. This was evidenced by a significant reduction of −24.4 points in the Clinician-Administered PTSD Scale for DSM-5 (CAPS-5, primary endpoint) scores, compared to −13.9 points in the placebo group (*p* < 0.0001, effect size *d* = 0.91). Similarly, functional impairment scores, measured by the Sheehan Disability Scale (SDS, secondary endpoint), decreased by −3.1 points in the MDMA group versus −2.0 points in the placebo group (*p* = 0.0116, effect size *d* = 0.43). Notably, 67% of MDMA-treated participants (28/42) no longer met the diagnostic criteria for PTSD by the end of the study, a success rate significantly higher than the 32% (12/37) observed in the placebo group. Moreover, 33% of participants in the MDMA group (14/42) achieved full remission after three treatment sessions, compared to only 5% (2/37) in the placebo group. This study finally demonstrated that MDMA therapy is not only effective, but also safe. Patients reported only mild and transient side effects (muscle rigidity, decreased appetite, nausea, hyperhidrosis, and feeling cold), with no significant increase in suicidality or cardiac toxicity [64]. In the subsequent MAPP2 trial, results confirmed the efficacy of the MDMA–psychotherapy protocol also in a more heterogeneous population (*n* = 104, 53 assigned to MDMA and 51 to placebo), including ethnic minorities and individuals with histories of complex trauma [6]. MDMA significantly attenuated PTSD symptomatology compared to the placebo, as measured by a reduction in the CAPS-5 total severity score from baseline to 18 weeks (−23.7 vs. −14.8, respectively). MDMA also significantly mitigated functional impairment with a reduction in the SDS from baseline (treatment difference: −1.20 (−2.26, −0.14); *p* = 0.03, *d* = 0.4). At the end of the study, 71.2% (*n* = 37/52) of patients in the MDMA group no longer had clinical symptoms of PTSD after treatment, compared to 47.6% (*n* = 20/42) in the placebo group. Only 9.4% (*n* = 5/53) of participants in the MDMA group reported relevant adverse events compared to 3.9% (*n* = 2/51) in the placebo group, with no deaths or serious events. No treatment-emergent adverse events (TEAEs) of MDMA abuse, misuse, physical dependence, or diversion were reported. MAPP2 has thus strengthened the evidence on the efficacy of MDMA as a potential “breakthrough therapy” for PTSD, also obtaining U.S. Food and Drug Administration (FDA) designation to facilitate the approval of the drug [6].

### 4.2. Completed Clinical Trials of Psilocybin in TRD and PTSD

Psilocybin has been studied primarily in the context of TRD, but, to date, only two phase 2 studies (COMP001 and COMP003) have completed data collection and made results available (Table 2). In the COMP001 trial (ClinicalTrials.gov identifier: NCT03775200), 233 patients with TRD received a dose of 25 mg (*n* = 79), 10 mg (*n* = 75), or 1 mg (control group, *n* = 79) of psilocybin, in a single session, with subsequent psychological support to complement the experience [66]. The primary endpoint was to assess the severity of depression by change in the Montgomery–Åsberg Depression Rating Scale (MADRS) score. The results indicated that the 25 mg group had a significantly greater reduction in the MADRS score than the control group (treatment difference: −6.6; 95% confidence interval [CI], −10.2 to −2.9; *p* < 0.001). At week 3, 37% (*n* = 29/79) of participants in the 25 mg group met response criteria (MADRS reduction ≥50%) and 29% (*n* = 23/79) achieved remission (MADRS ≤10), compared with 18% (*n* = 14/79) and 8% (*n* = 6/79), respectively, in the 1 mg group. At week 12, a sustained response was observed in 20% (*n* = 16/79) of the 25 mg group compared to 10% (*n* = 8/79) in the 1 mg group, although the results were not statistically conclusive. Adverse events occurred in 179 of 233 participants (77%) and included headache, nausea, and dizziness. Serious adverse events were more frequent in the 25 mg and 10 mg groups compared to the 1 mg group. Suicidal ideation or behavior or self-harm occurred in all dose groups, but these events were more frequent at the 25 mg dose [66].

The COMP003 study (ClinicalTrials.gov identifier: NCT04739865) explored the use of psilocybin as an add-on treatment for patients with treatment-resistant depression (TRD) who are concomitantly taking a selective serotonin reuptake inhibitor (SSRI). This approach was designed to assess the compatibility of psilocybin with existing antidepressant therapies, providing a treatment option for patients who cannot discontinue SSRIs for clinical reasons [65]. The study involved 19 participants who received a single dose of 25 mg psilocybin accompanied by psychological support before and after administration. The primary objective of the study was to observe the change in the MADRS score from baseline to week 3 after psilocybin administration. Secondary endpoints were safety, including TEAEs and responder and remitter rates at week 3, and change from baseline to week 3 in the Clinical Global Impression–Severity (CGI-S) score. The results indicated an improvement in depressive symptoms, with a clinically meaningful reduction in the MADRS score (−14.9; 95% CI, −20.7 to −9.2) at week 3. This improvement was evident at day 2 and was maintained throughout the 3-week follow-up. Additionally, 42.1% (*n* = 8/19) of participants met criteria for both response (MADRS reduction ≥ 50%) and remission (MADRS ≤ 10) at week 3, while the CGI-S score decreased by −1.3 points at week 3, indicating moderate improvement. Regarding safety, psilocybin was well tolerated, with 63.2% of participants experiencing TEAEs, the majority of which were mild and resolved on the same day. Common TEAEs included headache (31.6%) and increased blood pressure (15.8%). No serious adverse events or increases in suicidality were reported [65].

In addition to TRD, psilocybin is currently the subject of 10 studies for the treatment of PTSD. Among the most significant trials that have reached the completion phase with promising results is the COMP201 study (ClinicalTrials.gov identifier: NCT05312151) (Table 3). This phase 2 study focused on evaluating the safety and tolerability of psilocybin (25 mg) administered in a single dose to patients with PTSD [67]. The trial was conducted on a population of 22 participants and included a 12-week follow-up to monitor both the immediate and long-term effects of psilocybin administration, accompanied by psychotherapeutic support. The results, available on the study sponsor’s website, were promising, showing significant improvements in PTSD symptoms [67]. After 12 weeks, an average reduction of 29.5 points in the CAPS-5 scale and a reduction of 14.4 points in the SDS scale were observed. Treatment response was particularly significant: 81.8% (*n* = 18/22) of participants showed a reduction of at least 15 points in the CAPS-5 scale and 63.6% (*n* = 14/22) of participants achieved remission at week 4, maintaining high response and remission rates through week 12 (77.3% response and 54.5% remission). The study also met its primary safety endpoint: psilocybin was generally well tolerated, with no serious adverse events reported. The most common adverse events were mild and included transient symptoms such as headache (*n* = 11; 50.0%), nausea (*n* = 8; 36.4%), crying (*n* = 6; 27.3%), and fatigue (*n* = 6; 27.3%) [67].

### 4.3. Overview of Major Completed Clinical Trials of LSD in GAD

Studies on LSD are still limited. Among the most significant studies on LSD for the treatment of GAD is the recently completed MMED008 trial (ClinicalTrials.gov identifier: NCT05407064), a phase 2b trial that evaluated the safety and efficacy of escalating doses of LSD (lysergide tartrate, MM-120) in 198 patients with generalized anxiety [68] (Table 4). The aim of this study was to examine the efficacy of four doses of LSD tartrate (25 μg, 50 μg, 100 μg, and 200 μg) administered as a single dose and compared with a placebo group. Participants were divided into five groups, each of which received a different dose of LSD or a placebo, with follow-up and monitoring of anxiety symptoms using the Hamilton Anxiety Rating Scale (HAM-A). Preliminary results, available on the study sponsor’s website, showed that the group receiving 100 μg of LSD experienced a significant reduction in HAM-A scores, with a decrease of 21.3 points compared to 13.7 points in the placebo group already on the second day, and with a stable effect for four weeks [68]. This improvement was clinically and statistically significant, with a sustained effect up to the twelfth week, during which 65% (*n* = 129/198) of patients achieved a clinical response and 48% (*n* = 95/198) achieved the complete remission of symptoms. Treatment was generally well tolerated by participants. The most common side effects, which occurred mainly on the day of administration, included mild hallucinations, euphoria, temporary anxiety, headache, and nausea. These adverse events were mild to moderate in intensity and resolved spontaneously. No serious adverse effects were reported, and the study completion rate was high, with 97.5% (*n* = 193/198) of participants on the high dose completing the first four weeks. Based on the significant unmet medical need in the treatment of GAD, particularly in patients who do not respond to or tolerate currently available medications, together with initial clinical data from this phase 2b study, the FDA designated MM120 for GAD as a “breakthrough therapy” [69].

## 5. Conclusions

The rehabilitation of psychedelics in scientific and clinical research marks a pivotal moment in our understanding of consciousness, neuroplasticity, and psychiatric treatments. Evidence from recent trials demonstrates the strong potential of these substances to address complex and treatment-resistant mental health conditions, including PTSD, depression, and anxiety, with remarkable efficacy and safety under guided therapeutic frameworks. The outcomes not only achieve statistical significance, but in many cases, also provide tangible benefits in patients, with clinically relevant impacts for otherwise difficult-to-treat disorders. However, questions remain about the sustainability and durability of these therapeutic effects, underscoring the need for further studies in larger populations to consolidate the evidence, ensure long-term safety, determine the relevance of the psychological support, and define the integration of these approaches into clinical practice. Concurrently, advancements in neuroimaging and circuit-level analyses reveal how psychedelics modulate brain network dynamics, offering new insights into their mechanisms of action. The integration of frameworks like REBUS and CSTC has further deepened our understanding of the interplay between neural adaptation and subjective experiences.

However, challenges remain. The variability in individual responses, the role of subjective experience in clinical outcomes, and the influence of the set and setting highlight the need for personalized approaches to psychedelic therapy. Furthermore, significant gaps exist in linking molecular and circuit-level mechanisms to long-term behavioral and psychological outcomes, especially for what concerns humans, in whom we are limited mainly to functional analyses.

To fully realize the therapeutic and exploratory potential of psychedelics, interdisciplinary efforts are essential. Combining insights from neuroscience, psychiatry, pharmacology, and psychology, it would be possible to refine therapeutic protocols, optimize safety, and broaden their applications. Among all, the study of L5 pyramidal neurons advanced by psychedelics’ research has illuminated key neural mechanisms underlying consciousness. Together, they reveal how cortical synchronization and altered states shape self-perception, cognition, and sensory integration, offering a unified perspective on the structural, functional, and experiential dimensions of consciousness. The re-emergence of psychedelics is not just a rediscovery of ancient tools but a gateway to a deeper understanding of the human mind and its capacity for healing [70,71,72].

On this review, we decided to focus only on three compounds, namely LSD, psylocibin, and MDMA, as they are in the most advanced stages of clinical research. However, the landscape of psychedelics with therapeutic potential is vaster, comprising other drugs such as dimethyltryptamine (DMT)/ayahuasca, mescaline, ibogaine, and even dissociative compounds such as ketamine, which are currently being studied for the treatment of various disorders.

## 6. Future Perspectives

To fully harness the therapeutic potential of psychedelics for the treatment of many psychiatric disorders, future research should clarify the precise role of the set, setting, and psychotherapy, which create a safe and meaningful framework for the individual to process and integrate their experiences. By developing structured therapeutic protocols, improving public education, and investing in training and research, the field can move forward to integrating psychedelics into mainstream psychiatric care while minimizing risks and maximizing benefits [73]. Along the same lines, it would be interesting to clarify the contribution of mystical and psychedelic experiences to therapeutic outcomes, considering recent increasing evidence [74,75,76]. These experiences are thought to play a pivotal role in mediating the psychological and emotional transformations observed in psychedelic-assisted therapy. Understanding their precise impact could help optimize treatment protocols and better identify which individuals are most likely to benefit from such interventions.

In addition to this, the practice of microdosing is gaining increasing interest. Microdosing commonly consists in ingesting sub-perceptual doses of LSD, psylocibin, or other psychedelic drugs a few times per week for prolonged periods, at least several weeks or months. A future perspective could be to determine whether microdosing actually has positive effects, as suggested by some research, or if it is just a placebo [77,78]. Alongside the positive effects, however, researchers should take into account even the risks of this chronic low-dose administration of psychedelic substances, mainly related to organ toxicity such as cardiac fibrosis and valvulopathy [79].

Another question that must be addressed concerning psychedelics is whether they could be efficiently used as a powerful tool to analyze higher and complex brain functions, such as consciousness. According to this, L5 pyramidal neurons have been found to regulate conscious states through their role in synchronizing cortical activity, particularly under conditions such as anesthesia [71,80]. At the same time, psychedelics allow researchers to probe disruptions in self-perception, cognition, and sensory integration, offering unique insights into altered and hybrid states of consciousness. By integrating these findings, neuroscience is beginning to bridge the gap between structural, functional, and phenomenological dimensions of consciousness, paving the way for a more unified and comprehensive understanding of this elusive phenomenon [72].

## 7. Methods

This is a narrative review on the influence of psychedelics on brain connectivity systems and on their mechanism of action. Accordingly, we searched relevant keywords on PubMed, such as “psychedelics AND 5-HT2A”, “psychedelics AND mechanism of action”, “psychedelics AND (default mode network OR DMN)”, “psychedelics AND (Cortico-Striatal Thalamo-Cortical OR CSTC)”, and “psychedelics AND (Relaxed Beliefs Under Psychedel-ics OR REBUS)”.

An extensive search of the ClinicalTrials.gov database was conducted to identify clinical trials involving the use of MDMA, psilocybin, and LSD in the treatment of psychiatric disorders, such as PTSD, TRD, and GAD. The search was driven by the following targeted query strings: “MDMA or 3,4-methylenedioxymethamphetamine AND post-traumatic stress disorder”, “MDMA or 3,4-methylenedioxymethamphetamine AND treatment-resistant depression”, “MDMA or 3,4-methylenedioxymethamphetamine AND generalized anxiety disorder”, “psilocybin AND post-traumatic stress disorder”, “psilocybin AND treatment-resistant depression”, “psilocybin AND generalized anxiety disorder”, “LSD or lysergic acid diethylamide AND post-traumatic stress disorder”, “LSD or lysergic acid diethylamide AND treatment-resistant depression”, and “LSD or lysergic acid diethylamide AND generalized anxiety disorder”. Only completed clinical trials with available results, registered on ClinicalTrials.gov through 31 October 2024, were included in our analysis.

## Figures and Tables

**Figure 1 pharmaceuticals-18-00130-f001:**
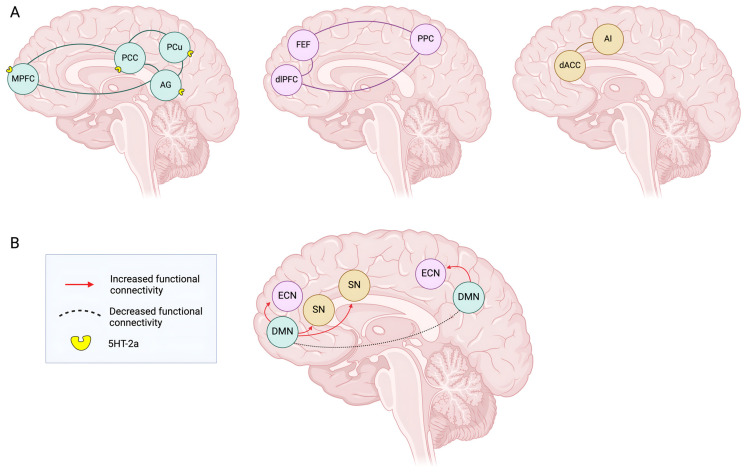
The psychedelic effect on the connectivity between the default mode network, executive control network, and salience network. (**A**) Key areas involved in DMN, ECN and SN networks. (**B**) Psychedelics’ assumption increases connectivity between DMN and SN and between DMN and ECN, together with a decreased connectivity within the hubs of the DMN. DMN: default mode network; ECN: executive control network; SN: salience network; AG: angular gyrus; AI: anterior insula; dACC: dorsal anterior cingulate cortex; dlPFC: dorsolateral prefrontal cortex; FEF: frontal eye field; MPFC: medial prefrontal cortex; PCu: precuneus; PCC: posterior cingulate cortex; PPC: posterior parietal cortex.

**Figure 2 pharmaceuticals-18-00130-f002:**
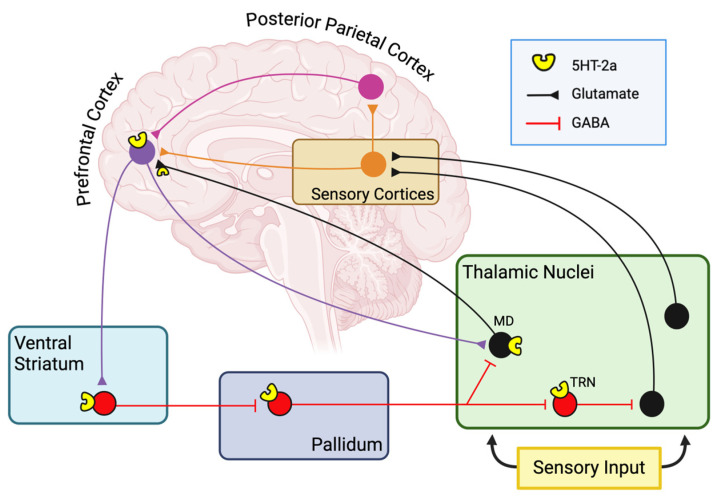
The psychedelic effect on the cortico-striatal thalamo-cortical (CSTC) circuitry. The CSTC circuit consists of the pyramidal neurons of the medial prefrontal layer V that project to the GABAergic neurons of the ventral striatum, which in turn inhibit specific GABAergic neurons of the pallidum that subsequently inhibit some thalamic nuclei that project back to the cortex. Each of these stations expresses 5-HT receptors, in particular 5-HT2AR. According to this scheme, it has been hypothesized that serotonergic psychedelics are able to reduce the effectiveness of thalamic gating by stimulating 5-HT2A receptors present at various levels of the circuit, resulting in the increase in the sensory perception and dissolution of the ego that occur in psychedelic states.

**Table 1 pharmaceuticals-18-00130-t001:** Major completed clinical trials investigating MDMA in PTSD (Up to October 2024).

Phase 3—Interventional Studies				
Reference ID	Date, Start–End	Aims	Arms and Interventions	Results
NCT03537014 (MAPP1)	Nov 2018–Aug 2020	To assess efficacy and safety of MDMA-assisted therapy (MDMA-AT) vs. placebo in participants with severe PTSD	▪ Experiment: 3 sessions (3–5 weeks apart) with 80/120 mg MDMA + 40/60 mg supplemental dose (1.5–2 h later), combined with therapy ▪ Placebo Comparator: 3 sessions (3–5 weeks apart) with inactive placebo, combined with therapy	▪ CAPS-5 score: −24.4 (MDMA-AT) vs. −13.9 (placebo), *p* < 0.0001, *d* = 0.91 ▪ SDS score: −3.1 (MDMA-AT) vs. −2.0 (placebo), *p* = 0.0116, *d* = 0.43 ▪ PTSD criteria: 67% (MDMA-AT) no longer met criteria vs. 32% (placebo); remission rates: 33% (MDMA-AT) vs. 5% (placebo) ▪ Safety: mild and transient side effects; no increased suicidality or cardiac toxicity in MDMA-AT
NCT04077437 (MAPP2)	Sep 2020–Nov 2022	To evaluate efficacy and safety of MDMA-AT vs. placebo in participants with moderate-to-severe PTSD	▪ Experiment: 80–120 mg MDMA + 40/60 mg supplemental dose (1.5–2 h later), combined with therapy ▪ Placebo Comparator: inactive placebo combined with therapy	▪ CAPS-5 score: −23.7 (MDMA-AT) vs. −14.8 (placebo); *p* < 0.001, *d* = 0.7 ▪ SDS score: −3.3 (MDMA-AT) vs. −2.1 (placebo); *p* = 0.03, *d* = 0.4 ▪ PTSD criteria: 71.2% (MDMA-AT) no longer met criteria vs. 47.6% (placebo) ▪ Safety: relevant adverse events in 9.4% (MDMA-AT) vs. 3.9% (placebo); no deaths or serious events

**Table 2 pharmaceuticals-18-00130-t002:** Major completed clinical trials investigating psilocybin in TRD (Up to October 2024).

Phase 2—Interventional Studies				
Reference ID	Date, Start–End	Aims	Arms and Interventions	Results
NCT03775200 (COMP001)	Mar 2019–Sep 2021	To evaluate safety and efficacy of psilocybin in TRD	▪ Experiment: Low-dose psilocybin (1 mg) ▪ Experiment: Medium-dose psilocybin (10 mg) ▪ Experiment: High-dose psilocybin (25 mg)	▪ MADRS score: −12.0 (25 mg), −7.9 (10 mg), and −5.4 (1 mg) at week 3; significant difference between 25 mg and 1 mg groups (*p* < 0.001) ▪ Week 3 response/remission rate: 37%/29% (25 mg) vs. 18%/8% (1 mg) ▪ Week 12 response rate: 20% (25 mg) vs. 10% (1 mg); not statistically conclusive ▪ Safety: 77% experienced adverse events; relevant events more frequent with 25 mg and 10 mg. Suicidal ideation/behavior occurred across all groups, highest in 25 mg
NCT04739865 (COMP003)	Aug 2020–Oct 2021	To assess safety and efficacy of psilocybin as adjunct therapy in TRD	▪ Experiment: 25 mg COMP360 psilocybin	▪ MADRS score: −14.9 at week 3 from a baseline of 31.7 ▪ CGI-S score: −1.3 at week 3, indicating moderate improvement ▪ Week 3 response/remission rate: 42.1% of participants met criteria for both response and remission ▪ Safety: 63.2% experienced TEAEs, mostly mild and resolved on same day; no serious events or increased suicidality

**Table 3 pharmaceuticals-18-00130-t003:** Major completed clinical trials investigating psilocybin in PTSD (Up to October 2024).

Phase 2—Interventional Studies				
Reference ID	Date, Start–End	Aims	Arms and Interventions	Results
NCT05312151 (COMP201)	Jun 2022–Feb 2024	To assess safety and tolerability of COMP360 in PTSD	▪ Experiment: 25 mg COMP360 psilocybin	▪ CAPS-5 score: −29.5 from baseline at week 12 ▪ SDS score: −14.4 from baseline at week 12 ▪ Week 4 response/remission rate: 81.8% response and 63.6% remission ▪ Week 12 response/remission rate: 77.3% response and 54.5% remission ▪ Safety: well tolerated; no serious adverse events

**Table 4 pharmaceuticals-18-00130-t004:** Major completed clinical trials investigating LSD in GAD (Up to October 2024).

Phase 2—Interventional Studies				
Reference ID	Date, Start–End	Aims	Arms and Interventions	Results
NCT05407064 (MMED008)	Aug 2022–Nov 2023	To evaluate effects of four MM-120 (LSD D-tartrate) doses in GAD	▪ Placebo Comparator: Arm 1—Placebo ▪ Experiment: Arm 2—25 μg MM-120 ▪ Experiment: Arm 3—50 μg MM-120 ▪ Experiment: Arm 4—100 μg MM-120 ▪ Experiment: Arm 5—200 μg MM-120	▪ HAM-A score: −21.3 points (100 µg) vs. −13.7 points (placebo); rapid, durable effect from day 2 to week 4 and sustained at week 12 ▪ Week 12 response/remission rate: 65% response and 48% remission (100 µg) ▪ Safety: well tolerated; mild/moderate adverse events on dosing day

## Data Availability

Not applicable.

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
