# Peer review of "Uncovering Psychedelics: From Neural Circuits to Therapeutic Applications"

_pharmaceuticals, 2025, doi:10.3390/ph18010130_

Round 1
Reviewer 1 Report
Comments and Suggestions for Authors
Psychedelics, known for their cultural significance, are emerging as therapeutic tools for conditions like PTSD, depression, and anxiety. They enhance brain plasticity by altering connectivity, reducing rigidity, and increasing functional flexibility. Although psychedelics have existed for a long time there is still to explore the neural mechanisms behind the acute and chronic effects of psychedelics. In this manuscript the authors have reviewed and synthesized current knowledge on the therapeutic potential of psychedelics, focusing on their effects on brain connectivity, neural plasticity, and psychiatric treatment. They discuss the mechanisms like the Default Mode Network (DMN) and REBUS model, highlighted findings from clinical trials of MDMA, psilocybin, and LSD for treating conditions like PTSD, depression, and anxiety, and gaps in understanding the link between psychedelics’ molecular mechanisms and clinical efficacy. Such review articles are important to the field and the authors have nicely summarized the published findings.
I would recommend some changes to this manuscript before it is deemed good for publication.
1) The authors should have a methodology section where they should describe the literate survey methods including the time window used to filter the references and the strings used to pull the relevant references
2) A similar section for the clinical trials to describe the search string of clinical trials and inclusion and exclusion criteria for including or excluding a clinical trial in this review if any.
3) I also suggest that the authors include a future directions section at the end of the manuscript, where they can share their perspectives and recommendations on the potential progression of the field. This section could outline key areas for future research and highlight the essential questions that must be addressed.
Author Response
Reviewer 1
Psychedelics, known for their cultural significance, are emerging as therapeutic tools for conditions like PTSD, depression, and anxiety. They enhance brain plasticity by altering connectivity, reducing rigidity, and increasing functional flexibility. Although psychedelics have existed for a long time there is still to explore the neural mechanisms behind the acute and chronic effects of psychedelics. In this manuscript the authors have reviewed and synthesized current knowledge on the therapeutic potential of psychedelics, focusing on their effects on brain connectivity, neural plasticity, and psychiatric treatment. They discuss the mechanisms like the Default Mode Network (DMN) and REBUS model, highlighted findings from clinical trials of MDMA, psilocybin, and LSD for treating conditions like PTSD, depression, and anxiety, and gaps in understanding the link between psychedelics’ molecular mechanisms and clinical efficacy. Such review articles are important to the field and the authors have nicely summarized the published findings. I would recommend some changes to this manuscript before it is deemed good for publication.
We thank the reviewer for his/her positive comments on our manuscript.
1) The authors should have a methodology section where they should describe the literate survey methods including the time window used to filter the references and the strings used to pull the relevant references
We appreciated the reviewer’s comment and accordingly we have added a short methodology section (PAR 7), explaining how we performed our research through the use of keywords. However, we would like to remind to the reviewer that this is not a systematic review, thus we did not use a full research string (as it would be for PRISMA guidelines). This is what we added, and on this topic we invite the reviewer also to see reply # 2.
“ This is a narrative review on the influence of psychedelics on brain connectivity systems and on their mechanism of action. Accordingly, we searched on PubMed relevant keywords, such as: “psychedelics AND 5-HT2A”, “psychedelics AND mechanism of action“, “psychedelics AND (default mode network OR DMN)”, “psychedelics AND (Cortico-Striatal Thalamo-Cortical OR CSTC)”, “psychedelics AND (Relaxed Beliefs Under Psychedel-ics OR REBUS)”.
2) A similar section for the clinical trials to describe the search string of clinical trials and inclusion and exclusion criteria for including or excluding a clinical trial in this review if any.
In relation to the clinical trials, in the methodology section we have added a brief explanation of how we conducted the research and on the inclusion criteria in the methodology section, as follows:
“An extensive search of the ClinicalTrials.gov database was conducted to identify clinical trials involving the use of MDMA, psilocybin, and LSD in the treatment of psychiatric disorders, such as PTSD, TRD, and GAD. The search was driven by the following targeted query strings: “MDMA or 3,4-methylenedioxymethamphetamine AND post-traumatic stress disorder”, “MDMA or 3,4-methylenedioxymethamphetamine AND treatment-resistant depression”, “MDMA or 3,4-methylenedioxymethamphetamine AND generalized anxiety disorder”, “psilocybin AND post-traumatic stress disorder”, “psilocybin AND treatment-resistant depression”, “psilocybin AND generalized anxiety disorder”, “LSD or lysergic acid diethylamide AND post-traumatic stress disorder”, “LSD or lysergic acid diethylamide AND treatment-resistant depression”, and “LSD or lysergic acid diethylamide AND generalized anxiety disorder.” Only completed clinical trials with available results, registered on ClinicalTrials.gov through October 31, 2024, were included in our analysis.”
3) I also suggest that the authors include a future directions section at the end of the manuscript, where they can share their perspectives and recommendations on the potential progression of the field. This section could outline key areas for future research and highlight the essential questions that must be addressed.
We thank the reviewer for this useful input, and we added a whole new paragraph 6 “Future perspectives”, adding potential developments and future challenges in this field (see in the text).

Reviewer 2 Report
Comments and Suggestions for Authors
I have read and reviewed the review by Melani et al entitled “Uncovering the psychedelic mind: from neural circuits to therapeutic applications”, which has been submitted for publication in Pharmaceuticals. The role of hallucinogens as recreational drugs and also potentially therapeutic agents is of continuing interest. The manuscript is generally well written and organized. Nonetheless, I have a few questions and concerns, which are detailed below, in the order in which they appear in the manuscript.
General Questions:
Integration of the 3 themes: The authors summarize research with psychedelics with respect to (1) molecular and cellular targets, (2) circuitries and brain networks theoretically involved and (3) large-scale clinical trials of psychedelics. What seems to be missing is a more comprehensive integration of these three areas. The paragraph on MDMA and PTSD (lines 212-222) is a good first step.
Clinical Study Endpoints - Qualitative Assessments: Can the authors provide a bit of context regarding the changes in clinical endpoints, i.e., how clinically relevant are the changes identified as significant? Are these really dramatic effects or only modest effects with statistical significance only?
Mysticism and its Potential Importance for Other Clinical Outcomes ? Would the authors consider the possibility that the psychedelic effect underlies benefits in patients with TRD, PTSD and anxiety? The authors might want to consider including some of the work by Roland Griffiths with regard to the ‘mysticism’ effects of psychedelic drugs (Psychopharmacology, 2006), and the possibility that the clinical benefits observed in TRD, PTSD or GAD may occur secondarily to this effect (Psychopharmacology, 2016)?
Limited almost exclusively to MDMA, LSD and psilocybin: The authors should note that the clinical studies reviewed are limited to a very small group of the potential drugs, namely MDMA, LSD and psilocybin. In the pharmaceutical industry, these three drugs might be considered as lead compounds, all needing some degree of improvement in terms of selectivity, bioavailability, etc, but that does not happen with the psychedelics. This is unfortunate, but certainly not the authors’ fault. But perhaps it warrants noting, especially in the journal Pharmaceuticals.
Writing-Specific Concerns:
(In the order in which they appear in the manuscript)
Title - Lines 1 & 2: ‘Uncovering the psychedelic mind’ sounds racy and cool, but the authors are really more precisely studying psychedelic drugs, not the mind. I would suggest something like “Psychedelic/Hallucinogenic drugs: Biological targets, neural circuits and therapeutic applications”.
Abstract, line 13: I would suggest “...as potential breakthrough ….”
Key Words, line 32: Please add TRD and GAD
Page 2, line 58: “...negative bias and social withdrawal….” Please include the reference
Page 2, lines 70-71: … understanding the changes in neural circuits underlying the neuroplastic effects of psychedelics is ….”
Page 2, lines 79-81: This sentence seems unnecessary, and possibly even gets in the way of connecting the paragraphs above and below.
Page 4, line 163: …discovery of more selective drugs….
Page 4, line 174: … demonstrated that MDMA exerts actions at serotonin….
Page 5, lines 197-108: Precisely WHAT behavioral effects of MDMA are produced by stimulation of 5-HT release in the Nucleus Accumbens? Please expound.
Page 5, lines 212-222: This seems to be an integration of molecular receptor studies and clinical work on PTSD; should it be moved to the PTSD clinical section?
Page 6, starting with line 272: there seems to be a problem with the left margin; it should be flush, and indented, correct?
In summary, this manuscript is an interesting read and does a pretty good job summarizing the state of affairs regarding psychedelic drugs and their potential for clinical benefits. I have only minor concerns, but no major reservations regarding publication of the manuscript in Pharmaceuticals.
See above
Author Response
Reviewer 2
I have read and reviewed the review by Melani et al entitled “Uncovering the psychedelic mind: from neural circuits to therapeutic applications”, which has been submitted for publication in Pharmaceuticals. The role of hallucinogens as recreational drugs and also potentially therapeutic agents is of continuing interest. The manuscript is generally well written and organized. Nonetheless, I have a few questions and concerns, which are detailed below, in the order in which they appear in the manuscript.
We thank the reviewer for his/her positive comments on our manuscript.
General Questions:
Integration of the 3 themes: The authors summarize research with psychedelics with respect to (1) molecular and cellular targets, (2) circuitries and brain networks theoretically involved and (3) large-scale clinical trials of psychedelics. What seems to be missing is a more comprehensive integration of these three areas. The paragraph on MDMA and PTSD (lines 212-222) is a good first step.
We thank the reviewer for this comment, and we added a relevant part at the end of PAR-3, which integrates with PAR-2 and PAR-4.
We are also glad that reviewer has noticed our efforts in the paragraph on MDMA and PTSD (lines 212-222) as a good first step to integrate the different parts. We would like also to mention that in other parts of the review, we tried to integrate these three parts (e.g. lines 251-257; lines 314-322; lines 361-365).
This is the part we added at the end of PAR3:
“The progressive elucidation of the neural circuits underlying psychedelics' action is pivotal for advancing more effective therapeutic applications. Many psychiatric conditions, such as depression, anxiety, PTSD, and SUDs, are often rooted in altered brain connectivity, and could be better understood and treated by investigating how psychedelics modulate these circuits[59,60]. A deeper understanding of these mechanisms may pave the way for targeted interventions that leverage psychedelics to restore or enhance functional connectivity in affected networks. For instance, research demonstrates that psychedelics promote neural plasticity through the activation of 5-HT2A receptors, facilitating dendritic growth and synaptic remodeling. These changes may help to restore disrupted neural networks in treatment-resistant depression [61]. Additionally, psilocybin and other psychedelics increase brain entropy and desynchronize neural activity, altering rigid patterns of connectivity often associated with disorders like obsessive-compulsive disorder (OCD) or SUDs [62,63]. The ability of psychedelics to enhance functional connectivity and break maladaptive patterns of brain activity may also explain their efficacy in alleviating PTSD by allowing patients to process traumatic memories in a more adaptive way [64].”
Clinical Study Endpoints - Qualitative Assessments: Can the authors provide a bit of context regarding the changes in clinical endpoints, i.e., how clinically relevant are the changes identified as significant? Are these really dramatic effects or only modest effects with statistical significance only?
In relation to these questions, we believe that the results of the clinical trials are promising and could represent a valid therapeutic alternative for several mental disorders. However, we also think that further studies are strongly needed to confirm these results, especially in the long-term. Accordingly, to clarify better the clinical relevance of the included studies, we added this part to our conclusions:
“The outcomes not only achieve statistical significance, but in many cases, also provide tangible benefits in patients, with clinically relevant impacts for otherwise difficult-to-treat disorders. However, questions remain about the sustainability and durability of these therapeutic effects, underscoring the need for further studies in larger populations to consolidate the evidence, ensure long-term safety, determine the relevance of the psychological support and define the integration of these approaches into clinical practice”.
Mysticism and its Potential Importance for Other Clinical Outcomes ? Would the authors consider the possibility that the psychedelic effect underlies benefits in patients with TRD, PTSD and anxiety? The authors might want to consider including some of the work by Roland Griffiths with regard to the ‘mysticism’ effects of psychedelic drugs (Psychopharmacology, 2006), and the possibility that the clinical benefits observed in TRD, PTSD or GAD may occur secondarily to this effect (Psychopharmacology, 2016)?
Following reviewer’s suggestion, in “future perspectives” we included this part as follows (adding the citations suggested):
“Along the same lines, it would be interesting to clarify the contribution of mystical and psychedelic experiences to therapeutic outcomes, considering recent increasing evidence [76–78]. These experiences are thought to play a pivotal role in mediating the psychological and emotional transformations observed in psychedelic-assisted therapy. Under-standing their precise impact could help optimize treatment protocols and better identify which individuals are most likely to benefit from such interventions.”
Limited almost exclusively to MDMA, LSD and psilocybin: The authors should note that the clinical studies reviewed are limited to a very small group of the potential drugs, namely MDMA, LSD and psilocybin. In the pharmaceutical industry, these three drugs might be considered as lead compounds, all needing some degree of improvement in terms of selectivity, bioavailability, etc, but that does not happen with the psychedelics. This is unfortunate, but certainly not the authors’ fault. But perhaps it warrants noting, especially in the journal Pharmaceuticals.
We understand this issue suggested by the reviewer, therefore in the conclusions we added this as a limitation of our review, as follows:
“On this review, we decided to focus only on three compounds, namely LSD, psylocibin, and MDMA, as they are in the most advanced stages of clinical research. However, the landscape of psychedelics with therapeutic potential is vaster, comprising other drugs such as dimethyltryptamine (DMT)/ayahuasca, mescaline, ibogaine, and even dissociative compounds such as ketamine, that are currently being studied for the treatment of various disorders.”
In relation to the role of psychedelics as lead compounds, we agree with the reviewer that, for the moment, there is no evidence if these lead compounds would bring to the discovery of other chemically-related new drugs. However, we decided not to include this sentence in the text for the moment, hoping that the reviewer agrees with us.
Writing-Specific Concerns:
(In the order in which they appear in the manuscript)
Title - Lines 1 & 2: ‘Uncovering the psychedelic mind’ sounds racy and cool, but the authors are really more precisely studying psychedelic drugs, not the mind. I would suggest something like “Psychedelic/Hallucinogenic drugs: Biological targets, neural circuits and therapeutic applications”.
We took advantage from the reviewer’s comment and now we propose this title:
Uncovering psychedelics: from neural circuits to therapeutic applications
Abstract, line 13: I would suggest “...as potential breakthrough ….”
We have added the word “potential”.
Key Words, line 32: Please add TRD and GAD
We have added the two keywords. Because of the limit of 10 keywords, we have removed “neural plasticity” and “depression”.
Page 2, line 58: “...negative bias and social withdrawal….” Please include the reference
We have added this reference: https://doi.org/10.3390/psychoactives3030026.
Page 2, lines 70-71: … understanding the changes in neural circuits underlying the neuroplastic effects of psychedelics is ….”
We have changed the phrase with the correct form as mentioned.
Page 2, lines 79-81: This sentence seems unnecessary, and possibly even gets in the way of connecting the paragraphs above and below.
We have removed the sentence as asked.
Page 4, line 163: …discovery of more selective drugs….
We have substituted “new” with “more”.
Page 4, line 174: … demonstrated that MDMA exerts actions at serotonin….
We have corrected the form.
Page 5, lines 197-108: Precisely WHAT behavioral effects of MDMA are produced by stimulation of 5-HT release in the Nucleus Accumbens? Please expound.
We have modified the sentence as follows:
“Then, by the means of TRAP (targeted recombination in active populations) technique and optogenetics, they were able to demonstrate that the stimulation of 5-HT release in the Nucleus Accumbens (NAc) alone is sufficient to reproduce the behavioral effects of MDMA; in particular, in a rodent model researchers have observed empathy-like behaviors (Rein et al., 2024).”
Page 5, lines 212-222: This seems to be an integration of molecular receptor studies and clinical work on PTSD; should it be moved to the PTSD clinical section?
The research mentioned in these lines is a preclinical study of a rodent model of PTSD, therefore we preferred keeping it in the mechanism section.
Page 6, starting with line 272: there seems to be a problem with the left margin; it should be flush, and indented, correct?
We have corrected the problem with the left margin.
In summary, this manuscript is an interesting read and does a pretty good job summarizing the state of affairs regarding psychedelic drugs and their potential for clinical benefits. I have only minor concerns, but no major reservations regarding publication of the manuscript in Pharmaceuticals.
We thank the reviewer for his/her positive opinion on our manuscript.
